

# The Tunka-Grande scintillation array: Current results

Anna L. Ivanova[1,2,*], I. Astapov[3], P. Bezyazeekov[2], E. Bonvech[4], A. Borodin[5],
N. Budnev[2], A. Bulan[4], D. Chernov[4], A. Chiavassa[6], A. Dyachok[2], A. Gafarov[2],
A. Garmash[1,7], V. Grebenyuk[5,8], E. Gress[2], O. Gress[2], T. Gress[2], A. Grinyuk[5], O. Grishin[2],
A. D. Ivanova[2], N. Kalmykov[4], V. Kindin[3], S. Kiryuhin[2], R. Kokoulin[3], K. Kompaniets[3],
E. Korosteleva[3], V. Kozhin[3], E. Kravchenko[1,7], A. Kryukov[4], L. Kuzmichev[4,2], A. Lagutin[9],
M. Lavrova[5], Y. Lemeshev[2], B. Lubsandorzhiev[10,4], N. Lubsandorzhiev[4], A. Lukanov[10],
D. Lukyantsev[2], S. Malakhov[2], R. Mirgazov[2], R. Monkhoev[2], E. Osipova[4],
A. Pakhorukov[2], L. Pankov[2], A. Pan[5], A. Panov[4], A. Petrukhin[3], I. Poddubnyi[2],
D. Podgrudkov[4], V. Poleschuk[2], V. Ponomareva[2], E. Popova[4], E. Postnikov[4], V. Prosin[4],
V. Ptuskin[11], A. Pushnin[2], R. Raikin[9], A. Razumov[4], G. Rubtsov[10], E. Ryabov[2],
Y. Sagan[5,8], V. Samoliga[2], A. Satyshev[5], A. Silaev[4], A. Silaev (junior)[4], A. Sidorenkov[10],
A. Skurikhin[4], A. Sokolov[1], L. Sveshnikova[4], V. Tabolenko[2], L. Tkachev[5,8], A. Tanaev[2],
M. Ternovoy[2], R. Togoo[12], N. Ushakov[10], A. Vaidyanathan[1], P. Volchugov[4], N. Volkov[9],
D. Voronin[10], A. Zagorodnikov[2], D. Zhurov[2,13] and I. Yashin[3]

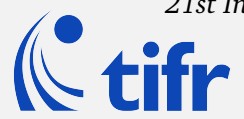

## Abstract

The Tunka-Grande experiment is a scintillation array with about 0.5 km$^2$ sensitive area at Tunka Valley, Siberia, for measuring charged particles and muons in extensive air showers (EASs). Tunka-Grande is optimized for cosmic ray studies in the energy range 10 PeV to about 1 EeV, where exploring the composition is of fundamental importance for understanding the transition from galactic to extragalactic origin of cosmic rays. This paper attempts to provide a synopsis of the current results of the experiment. In particular, the reconstruction of the all-particle energy spectrum in the range of 10 PeV to 1 EeV based on experimental data from four observation seasons is presented.

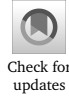

**1** Novosibirsk State University, Pirogova str. 2, Novosibirsk, 630090 Russia
**2** Irkutsk State University, Karl Marx str. 1, Irkutsk, 664003 Russia
**3** National Research Nuclear University MEPhI (Moscow Engineering Physics Institute),
Kashirskoe hwy 31, Moscow, 115409 Russia
**4** Lomonosov Moscow State University, Skobeltsyn Institute of Nuclear Physics,
Leninskie gory 1 (2), GSP-1, Moscow, 119991 Russia
**5** Joint Institute for Nuclear Research, Joliot-Curie str. 6, Dubna, 141980 Russia
**6** Dipartimento di Fisica Generale Universiteta di Torino and INFN,
Via P. Giuria 1, Turin, 10125 Italy



**7** Budker Institute of Nuclear Physics SB RAS,
Ac. Lavrentiev Avenue 11, Novosibirsk, 630090 Russia
**8** Dubna State University, Universitetskaya str. 19, Dubna, 141982 Russia
**9** Altai State University, pr. Lenina 61, Barnaul, 656049 Russia
**10** Institute for Nuclear Research RAS,
prospekt 60-letiya Oktyabrya 7a, Moscow, 117312 Russia
**11** IZMIRAN, Kaluzhskoe hwy 4, Troitsk, Moscow, 4108840 Russia
**12** Institute of Physics and Technology Mongolian Academy of Sciences,
Enkhtaivan av. 54B, Ulaanbaatar, 210651 Mongolia
**13** Irkutsk National Research Technical University, Lermontov str. 83, Irkutsk, 664074 Russia

⋆ annaiv.86@mail.ru

## Contents

## 1 Introduction

An important task of modern astrophysics is to study cosmic rays in the energy range from 10 PeV to about 1 EeV, where exploring the composition is of fundamental importance for understanding the transition from galactic to extragalactic origin of cosmic rays. Studies in this energy range are also interesting from the point of view of search for gamma rays with energy more than 100 TeV.

The Tunka-Grande scintillation facility described in reference [1] conducts scientific research in the fields of high-energy gamma-ray astronomy and cosmic ray physics by detecting the charged and muon components of EAS. It is located in Tunka Valley, 50 km of the Lake Baikal and together with four other independent arrays (TAIGA-HiSCORE and Tunka-133 Cherenkov arrays, TAIGA-Muon scintillation array, and network of Imaging Atmospheric Cherenkov Telescopes TAIGA-IACT) forms the Tunka experimental complex.

## 2 Five seasons of Tunka-Grande scintillation array operation

During the four seasons from 2017 to 2021, there were 691 days of Tunka-Grande operation. The array trigger condition was a coincidence of any three surface detectors within 5 μs. During this period, about 3,409,000 triggering events were detected on the Tunka-Grande area over 9100 h of operation. The mean trigger rate is about 0.1 Hz.

Only successfully reconstructed events with a zenith angle $\theta \leq 35^o$ and core position in a circle around the centre of the array with a radius R < 350 m were selected. The threshold energy of 100 persent registration efficiency for chosen area and zenith angles is 10 PeV. The number of selected events with energies above 10 PeV was about 240,000. Approximately 2000 events from them had energies above 100 PeV.

In the first observation season from 2016 to 2017, Tunka-Grande operated by Tunka-133 trigger. There were 475 h of joint operation and about 77,000 events.

The data from the first season were used to analyze joint events and assess the quality of the Tunka-Grande reconstruction technique. Unlike the weather-independent Tunka-Grande, only clear moonless nights are used for the correct events reconstruction by Tunka-133 Cherenkov array. This condition reduced the time suitable for joint events analysis to 357 hours and the number of joint events to 25,000. About 6000 events of them had core position in the circle with radius of 350 m, zenith angle of up to $35^o$ and energy above 10 PeV.

## 3 EAS and CR parameters reconstruction technique

The EAS and CR parameters reconstruction procedure includes several stages.

At the first stage, EAS pulses main parameters (amplitude, pulse area, registration time) are determined, calibration procedures are carried out, and the most probable energy deposit per charged particle, necessary for further reconstruction of the particle density in the detectors of the array, is determined. Then density of particles in ground and underground detectors, the initial values of the core position, the arrival direction of the EAS, the total number of particles of the electron-photon and muon components are calculated.

The shower arrival direction is determined by fitting the measured pulse front delay using the curved shower front formula, which was obtained at the KASCADE-Grande experiment [2].

An important element of the EAS and CR parameters reconstruction procedure is the function of the lateral distribution of particles (LDF). The lateral distribution of charged particles is described using the specific EAS-MSU function [3]. Greisen function [3] is used for muons.

Next, the shower core coordinates, number of muons and charged particles, and slope of the LDF are refined by minimizing the functional using independent variables. In addition, the age of the shower and the density of charged particles and muons at a distance of 200 m from EAS core position are determined.

As a measure of energy, we use the charged particles density at a core distance of 200 m – $\rho_{200}$ parameter rescaled relative to the measured zenith angle $\theta$ for atmospheric depth from sea level for the Tunka Valley $x_0 = 960 g/sm^2$ and obtained from experimental data average value of absorption path length $\lambda = 260 g/sm^2 : \rho_{200}(\theta = 0)$ [1]. The $\rho_{200}(\theta = 0)$ conversion to energy is carried out according to formula:

$$E_0 = 10^{15.99} \cdot (\rho_{200}(\theta = 0))^{0.84} . \tag{1}$$

Correlation $\rho_{200}(\theta = 0)$ with primary energy was determined using the experimental results of the Tunka-133 Cherenkov array. The $\rho_{200}(\theta = 0)$ value was calculated from the data of the Tunka-Grande scintillation array, and the energy value was taken from the data of the

Tunka-133 Cherenkov array. This technique is based on a comparison of experimental data and practically does not depend on the hadron interaction model.

## 4 Estimating the accuracy of the EAS and CR parameters reconstruction

The Tunka-133 Cherenkov array [4] demonstrate good performance and high reliability of equipment. Therefore, to determine the accuracy of the reconstruction of the EAS parameters measured with the Tunka-Grande array, the EAS parameters reconstructed from Tunka-133 experimental data are used.

The search for joint events was performed within the time range of minus plus 10 microseconds in showers with zenith angles of up to $35^o$, detected in a circle with $R < 350$ m.

The average difference between Tunka-Grande and Tunka-133 zenith angles doesn't exceed than $0.03^o$, for azimuth angles it is also less than $0.03^o$, the standard deviations are $\sigma_\theta, \sigma_\phi \approx 1.3^o$. The average differences between Tunka-Grande and Tunka-133 shower core coordinates don't exceed 2 meters, the standard deviation is $\sigma = 16.4$ m for X-coordinate and is $\sigma = 17$ m for Y-coordinate.

The distribution of the difference between the EAS arrival directions and distribution of the difference between the EAS core positions measured by the Tunka-Grande and by the Tunka-133 are shown in the Fig. 1.

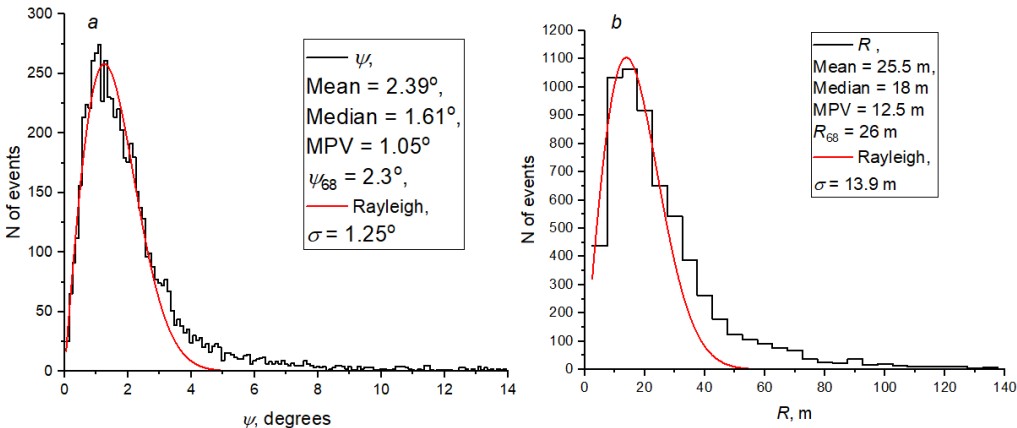

Figure 1: The accuracy of the arrival direction (*a*) and core position (*b*) reconstruction by the Tunka-Grande array in comparison with data of Tunka-133 array [4].

The thresholds for parameters $\psi$ and $R$ on the 68% - confidence level, within which 68% events are located,were taken by us as angular resolution and core position resolution of the Tunka-Grande array, respectively. The $\psi_{68}$ is equal to $2.3^o$ and $R_{68}$ is equal to 26 m.

The energy resolution is 36% for events with $E \geq 10$ PeV. It becomes better with increasing energy and doesn't exceed 15% for energies above 100 PeV.

## 5 Energy spectrum

To reconstruct the differential energy spectrum the number of events inside each bin of 0.1 in $\lg E_0$ was calculated. The differential uncorrected intensity is obtained by dividing this number by the selected effective area, the selected effective solid angle, the observation time, and the

energy bin width. We have selected events with zenit angle $\theta \leq 35^o$ and core position in a circle with a radius of 350 meters.

The spectrum of Tunka-Grande (Fig. 2, $a$) shows several features, deviating from a single power law. At an energy of about 20 PeV, the power law index changes from $\gamma = 3.18$ to $\gamma = 3.00$. And the spectrum becomes much steeper with $\gamma = 3.25$ above 100 PeV.

The spectrum is compared with the results from Tunka-133 [4], KASCADE-Grande [5], TALE [6] and Ice Top [7] facilities (Fig. 2, $b$). It demonstrates good agreement with data of large terrestrial facilities.

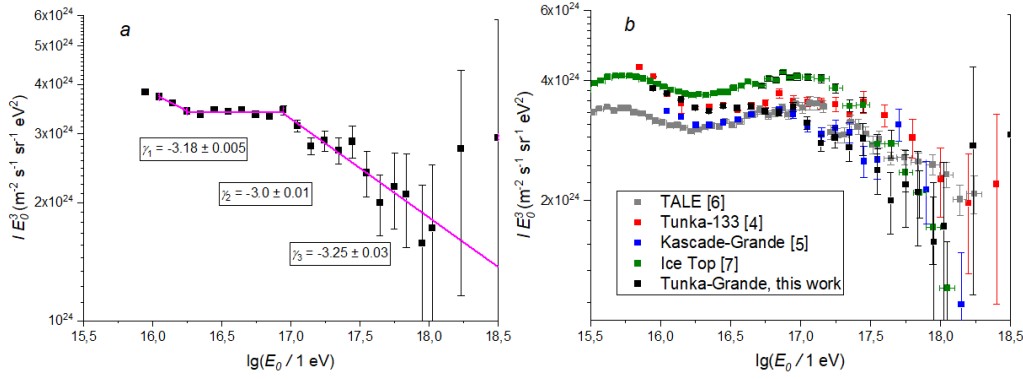

Figure 2: Differential primary cosmic-ray energy spectrum, $a$ - with a fit of a doubly broken power law, $b$ - comparison to other experimental results. The vertical lines show the value of the statistical errors.

## 6  Gamma-hadron discrimination

Nowadays the usual technique for searches for primary $\gamma$ in EASs is to discriminate gamma-ray primaries from the hadronic background by identifying muon-poor or even muon-less EASs. For this a computer simulation of the Tunka-Grande detectors operation [8,9] was done with help of CORSIKA(QGSJET-II-04,GHEISHA interaction models) and Geant4 softwares.

The Fig. 3, $a$ shows the distribution of the muon number in underground Tunka-Grande detectors versus $\rho_{200}$ parameter. Blue circles - experiment data ($N_\mu$ and $\rho_{200}$ are calculated from EAS events detected by Tunka-Grande array). Green circles - data of simulated Eass, initiated by gamma quantum. The events without any detected muons are plotted with $lg(N_\mu) = -1$ to be visible at the logarithmic axis. Lower red line indicates the selection criteria. There is no excess of events consistent with a gamma-ray signal seen in the data. Hence, we assume that all events below the selection line are primary $\gamma$-rays and set upper limits on the gamma-ray fraction of the cosmic rays. An upper limit on the fraction of the $\gamma$ ray with respect to the cosmic ray flux was determined by formula with 90% C.L.(Feldman – Cousins) $N_{90} = 2.44$, $\epsilon_\gamma \sim 0.5$:

$$\frac{I_\gamma}{I_{CR}} < \frac{N_{90}}{N_{tot}\epsilon_\gamma}\left(\frac{E_{CR}}{E_\gamma}\right)^{-\beta+1}. \tag{2}$$

Fig. 3, $b$ displays the measurements on the gamma-ray fraction as a function of the energy, including this work, for the energy range of 10 PeV up to 100 EeV.

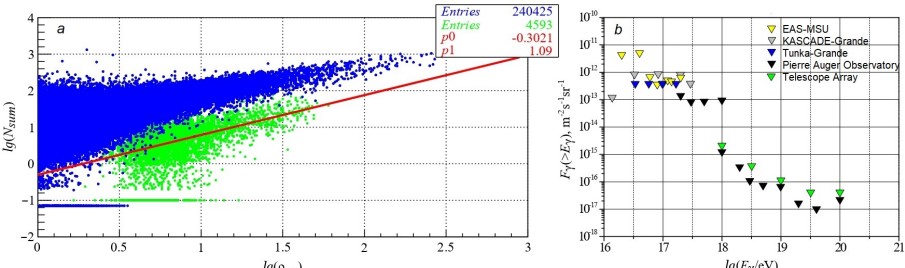

Figure 3: *a* - Scatter plot of the number of muons in underground detectors vs. $\rho_{200}$ parameter. Green circles - simulated EASs from $\gamma$, blue circles - experiment, red line - selection criteria. *b* - Gamma-hadron discrimination. Yellow $\nabla$ - EAS MSU [10], grey $\nabla$ - KASCADE Grande [11], blue $\nabla$ - this work, black $\nabla$ - PAO [12], green $\nabla$ - TA [13].

## 7 Conclusion

As a result of the "Tunka-Grande-Tunka-133" joint events analysis, it was found that for energies above 10 PeV, the angular resolution of Tunka-Grande array is $\psi_{68} \leq 2.3^o$, the core position resolution - $R_{68} \leq 26$ m, the energy resolution - $E_{68} \leq 36\%$ .

Based on 4 measurement seasons All-particle energy spectrum was reconstructed and the 90% C.L. upper limits to the diffuse flux of ultrahigh energy gamma rays for energies above 30 PeV were determined.

## Acknowledgements

**Funding information** The work was performed at the UNU "Astrophysical Complex of MSU-ISU" (agreement EB 075-15-2021-675). The work is supported by RFBR (grants 19-52-44002, 19-32-60003), the RSF(grants 19-72-20067(Section 2)), the Russian Federation Ministry of Science and High Education (projects FZZE-2020-0017, FZZE-2020-0024, FSUS-2020-0039, and FZZE-2022-0001).

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
