# Peer review of "The Tunka-Grande scintillation array: current results"

_SciPost Physics Proceedings, doi:SciPost Phys. Proc. 13, 011 (2023)_

## Round 1 · Referee Report · Anonymous (Referee 1) · 2022-9-4

Report
The paper gives a summary of results of air-shower measurements of the scintillator array of the Tunka installation .
The content is fitting the conference and the paper is valid for publication in this journal.
What follows are some comments and remarks to improve the readability of the manuscript.
Requested changes
abstract:
- please use latex km^2 instead of sq.km
section 1:
- It sound a bit strange to say "The important task..." I would suggest to use " An important task..."
- first line: ...is to study cosmic rays in the ....
- "....astronomy and cosmic-ray physics....the charged and the muon component of EAS."
- for a better understanding I suggest to include a few sentences for describing the array: size, sensitive area, shielded, not shielded, etc...
section 2:
- 3.409.000
- R<350m.... around what?
- percent
section 3:
- ...was obtained at the KASCADE-Grande....
- Greisen
- eqn(1): what is rho_200 (0) . what do you mean with this 0?
- laset sentence: can you shortly explained how you do the calibration and why it is not based on interaction models?
section 4:
- ...exceed 2 meters...
- Fig1: the distributions are by a folding from the uncertainties of Tunka-133 and Tunka-Grande. Therefore one should give here the information of the uncertainties by Tunka-133.
section 5:
- fig2: I assume the error bars are of pure statistical nature. This should be mentioned in teh figure caption.
section 6:
- the sentence "The Fig. 3, a shows the distribution of the muon number in underground Tunka-Grande detectors versus rho_200 parameter for measured showers by Tunka-Grande with simulated showers, induced gamma." is unclear. How you have the simulated muon number for measured showers? I assume all is simualted? What is the input energy spectrum and compostion in the blue data as well as for gamma simulations?
Author: Anna Ivanova on 2022-09-07 [id 2793]
(in reply to Report 1 on 2022-09-04)abstract: - please use latex km^2 instead of sq.km -
We agree
section 1:
It sound a bit strange to say "The important task..." I would suggest to use " An important task..." We agree
first line: ...is to study cosmic rays in the .... We agree
"....astronomy and cosmic-ray physics It seems to us that the generally accepted designation (at least the most common one) is cosmic ray physics, and not cosmic-ray physics ....the charged and the muon component of EAS. We agree
for a better understanding I suggest to include a few sentences for describing the array: size, sensitive area, shielded, not shielded, etc... ***"It contains 19 scintillation stations located on the Tunka-133 Cherenkov array area in a circle with a radius of about 400 m. Each station is composed of two parts: one on the Earth’s surface and one underground. The surface part, which consists of 12 counters, covers a total area of about 8 m^2 and detects EAS charged particles at the level of the array. The underground part which consists of 8 counters with a total area of about 5 m^2, is located under a 1.5 m layer of soil in the immediate vicinity of the surface part and is designed to separate the muon component of EAS." - The description of the installation is given in [1]. The text of the article has been reduced as much as possible to fit into the required 6 pages. If we insert a description of the installation, the number of pages is more than 6. ***
section 2: - 3.409.000 *** 3 409 000 ? - R<350m.... around what? *** It implies a circle with the center coinciding with the center of the array. "...in a circle around center of the array with a radius R <350m..." or "...in a circle with radius R<350m and center coinciding with the center of the array..." ? *** - percent *** ? To compare statistics in the field of high energies in different experiments, it is convenient to give the number of events, not percentages. The number of events is important and its conversion into percentages will complicate the comparison procedure for researchers.
section 3: - ...was obtained at the KASCADE-Grande.... We agree
Greisen Sorry. Misspell. Changes have been made
eqn(1): what is rho_200 (0) . what do you mean with this 0? *** rho_{200}(0) = rho_{200}(theta) * exp((x_0/lambda)(sec(theta) - 1 )), rho_{200}(0) - is rho_{200}(theta) recalculated to vertical (zenith angle theta = 0)**
laset sentence: can you shortly explained how you do the calibration and why it is not based on interaction models? *** Unfortunately, it was stated that the maximum possible size of the article is 6 pages. The description of the calibration will lead to a significant increase in the volume of the article. The energy reconstruction technique used depends less on interaction models than the standard one because the energy dependence on the ro200 parameter was obtained not from simulation data, but experimentally from joint events with Cherenkov installations. 1. The value of the parameter ro200 was calculated from the data of the scintillation array, and the energy value was taken from the data of the Cherenkov array Tunka-133. 2. The verification of the obtained dependence was carried out by analyzing the joint Tunka-Grande - TAIGA-HiSCORE events.***
section 4: - ...exceed 2 meters... We agree
section 5: - fig2: I assume the error bars are of pure statistical nature. This should be mentioned in teh figure caption. The vertical lines show the value of the statistical errors. We added this in the caption to the figure.
section 6: - the sentence "The Fig. 3, a shows the distribution of the muon number in underground Tunka-Grande detectors versus rho_200 parameter for measured showers by Tunka-Grande with simulated showers, induced gamma." is unclear. How you have the simulated muon number for measured showers? *** For measured showers we have number of muons, registered in each underground detector, and calculate sum: Nmu = Nmu1 + ... Nmu19. For simulated showers we do the same using simulation data.***
I assume all is simualted? *** NO. Green circles - simulated, Blue - Experiment***
What is the input energy spectrum and compostion in the blue data as well as for gamma simulations? Nowadays one of the most promising approaches to separate events from primary gamma rays from the charged cosmic rays background is the studying the muon EAS component, since the number of muons in a shower generated by a gamma quantum is an order of magnitude less than in the hadronic shower. This in turn requires Monte Carlo simulation of EASes, along with selecting and comparing experimental data. For more accurately assess the promise of Tunka-Grande array two-step computer simulation of the detectors operation was done. At the first, the development of an EAS is simulated, at the second, the detectors response to passage of elementary particles is simulated as well [https://inspirehep.net/files/aa0234bdbdb0fba04ac71a2c66532375]. To solve these tasks, the CORSIKA (Version 7.7401) and Geant4 packages were chosen as the software. EASes were generated from various primary particles (gamma quanta, protons, and iron nuclei) for energy range 16.5 < lg(E/eV) < 17.5 and for interval of the zenith angle 0° - 45°. Hadron interactions at low energies were calculated using the GHEISHA model, high-energy interactions were processed with the QGSJET-II-04 model. Simulation of the detectors response using Geant4 package is described in the report [https://iopscience.iop.org/article/10.1088/1742-6596/2103/1/012001/pdf]. The distributions of the muon number versus the ro200 ( that is, energy of primary particles) are shown in Figure 3a. The figure presents the sum of detected particles for underground parts of all 19 stations for each EAS. Blue circles - experiment data (muon number and ro200 are calculated from EAS events detected by Tunka-Grande array). green circles -data of simulated Eass, initiated by gamma quantum. The events without any detected muons are plotted with lg(Nm) = -1 to be visible at the logarithmic axis. It follows from the Figure 3a that, according to Tunka-Grande experimental data, it is possible to separate gamma-ray candidates from the charged cosmic rays background with an efficiency of no worse than 50 %. We can add references to articles [https://inspirehep.net/files/aa0234bdbdb0fba04ac71a2c66532375] and [https://iopscience.iop.org/article/10.1088/1742-6596/2103/1/012001/pdf].
*** Thank you for your comments!***

---

## Round 2 · List of Changes

abstract:
sq.km --> latex km^2

section 1:
"The important task..." --> " An important task..."
"... is cosmic ray studies..." --> "...is to study cosmic rays..."
"The Tunka-Grande scintillation facility [1] ..." --> "The Tunka-Grande scintillation facility described in reference [1] ..."
"...the charged and muon EAS components." --> "....the charged and the muon components of EAS."

section 2:
3409000 --> 3,409,000
240000 --> 240,000
77000 --> 77,000
25000 --> 25,000
"... in a circle with a radius R < 350 m..." --> "...in a circle around the centre of the array with a radius R < 350 m..."

section 3:
"...was obtained in a KASCADE-Grande..." --> " ...was obtained at the KASCADE-Grande...."
"Greysen" --> "Greisen"
"ρ200(0) " --> "ρ200(theta=0)"
"Correlation ρ200(0) with the primary energy is determined using the experimental results of
Tunka-133 Cherenkov array. This technique is based on ..." --> "Correlation ρ200(θ = 0) with primary energy was determined using the experimental results of the Tunka-133 Cherenkov array. The ρ200(θ = 0) value was calculated from the data of the Tunka-Grande scintillation array, and the energy value was taken from the data of the Tunka-133 Cherenkov array. This technique is based on ..."

section 4:
"...exceed than 2 meters..." --> " ...exceed 2 meters..."
add ref [4] in the caption of fig 1

section 5:
Add "The vertical lines show the value of the statistical errors." in the caption of fig 2

section 6:
"The Fig. 3, a shows the distribution of the muon number in underground Tunka-Grande detectors versus ρ200 parameter for measured showers by Tunka-Grande with simulated showers,
induced γ. Lower red line indicates the selection criteria." --> "The Fig. 3, a shows the distribution of the muon number in underground Tunka-Grande detectors versus ρ200 parameter. Blue circles - experiment data (Nµ and ρ200 are calculated from EAS events detected by Tunka-Grande array). Green circles - data of simulated Eass, initiated by gamma quantum. The events without any detected muons are plotted with l g(Nµ) = −1 to be
visible at the logarithmic axis. Lower red line indicates the selection criteria. "
Fig 3 caption"black circles" --> "blue circles"
+ 2 references
([8] M. Ternovoy et al, Simulation of the Tunka-Grande, TAIGA-Muon and TAIGA-HiSCORE
arrays for a search of astrophysical gamma quanta with energy above 100 TeV, J. Phys.
Conf. Ser. 1847, 012047 (2021), doi:10.1088/1742-6596/1847/1/012047.
[9] R.D. Monkhoev et al, Geant4 simulation of the Tunka-Grande experiment, J. Phys. Conf.
Ser. 2103, 012001 (2021), doi:10.1088/1742-6596/2103/1/012001.)
" For this a computer simulation of the Tunka-Grande detectors operation..." --> " For
this a computer simulation of the Tunka-Grande detectors operation [8,9]..."

---

## Editorial Decision

published